# Olfactory memory representations are stored in the anterior olfactory nucleus

Afif J. Aqrabawi [1,2] & Jun Chul Kim[1,3]

The anterior olfactory nucleus (AON) is the initial recipient of odour information from the olfactory bulb, and the target of dense innervation conveying spatiotemporal cues from the hippocampus. We hypothesized that the AON detects the coincidence of these inputs, generating patterns of activity reflective of episodic odour engrams. Using activity-dependent tagging combined with neural manipulation techniques, we reveal that contextually-relevant odour engrams are stored within the AON and that their activity is necessary and sufficient for the behavioural expression of odour memory. Our findings offer a new model for studying the mechanisms underlying memory representations.

[1] Department of Cell and Systems Biology, University of Toronto, Toronto M5S 3G5, Canada. [2] Picower Institute for Learning and Memory, Massachusetts Institute of Technology, Cambridge, MA 02139, USA. [3] Department of Psychology, University of Toronto, Toronto, ON M5S 3G3, Canada. ✉email: aqrabawi@mit.edu; kim@psych.utoronto.ca

As animals experience their environment, salient information is stored as a persistent physical change within the brain[1,2]. Ensembles of active neurons during encoding, or engrams, form the pillars of this memory substrate via the strengthening of synaptic connections[3–5]. Engrams can be later recalled when their activity is triggered by external cues, informing the animal on behaviourally relevant stimuli[6,7]. Identifying engrams is challenging due to their sparse, distributed nature and the previous lack of sophisticated tools. Recent studies have been more successful at observing, erasing and expressing several components of the engram in widespread cortical regions[6–21]. However, demonstrating an engram's content by recapitulating neural activity has been limited by the deficiency of current paradigms to produce an unambiguous behavioural output.

Most organisms are guided by olfaction, a highly conserved sensory system which can serve as a tractable model for investigating cognitive functions[22,23]. Centrally situated within the olfactory system is a ring-like cortical structure known as the anterior olfactory nucleus (AON). Anatomical investigations have characterised an extensive connectivity between the AON and neighbouring olfactory regions, yet its function remains largely elusive[24]. Previously, we demonstrated that odour perception can be altered by AON activity modulation[25]. We also found that the AON receives dense, topographically organised projections from the hippocampus, a structure highly implicated in navigation and episodic memory[26]. Furthermore, we determined the necessity of hippocampal-AON projections for recollecting spatiotemporal aspects of odour memory and identified a coincidence detection property for the AON[27]. Here, we combine a tamoxifen-inducible genetic tagging system with chemo- and optogenetic tools to tag and manipulate populations of neurons within the AON which constitute unique odour engrams. Our results establish the AON as a cortical repository for odour memory representations.

## Results

**AON neurons code coincident olfactory and contextual input.** We employed the $Fos^{CreER}$ mouse line which expresses Cre-recombinase downstream the c-Fos promoter and allows genetic labelling of active neurons following systemic injections of 4-hydroxytamoxifen (4-OHT)[28]. $Fos^{CreER}$ mice were infused with an adeno-associated virus (AAV) vector carrying a Cre-dependent reporter (AAV9-CAG-FLEX-GFP) into the AON. Following recovery, animals were injected with 4-OHT (50 mg/kg) and subsequently exposed to a salient odour and context, separately or in combination. We observed a high density of GFP-labelled cells within the AON in animals exposed to an odour-context pairing, yet neither stimulus alone increased the number of labelled cells compared to homecage and vehicle-only conditions (Supplementary Fig. 1a, b). Importantly, while c-Fos is commonly used as a proxy for neuronal activity, it may not exactly correlate with firing rates measured using various electrophysiological recordings. Thus, our data likely underestimates the number of neurons firing in response to the different behavioural conditions and does not dismiss the existence of 'odour-only' responsive cells within the AON. Nonetheless, an increase in c-Fos expression was distinctively observed in the piriform cortex, regardless of whether odour presentation occurred in the homecage or in a salient context. The necessity of contextually-relevant stimuli for c-Fos induction in the AON is evidence of a unique role in contextual processing beyond simply coding odour identity. These results confirm previous reports of coincidence detection at the AON and validate our tagging approach[27].

**AON engram activity is necessary for memory expression.** We aimed to determine whether silencing AON neurons, tagged in response to an odour-context pairing, can influence subsequent context-driven odour recollection. To this end, $Fos^{CreER}$ mice were bilaterally infused with an AAV vector carrying a Cre-dependent, inhibitory hM4D receptor gene (AAV8-hSyn-FLEX-hM4D-mCherry) or the reporter only (AAV8-hSyn-FLEX-mCherry) (Fig. 1a). Injections of 4-OHT produced high levels of hM4D-mCherry expression restricted to the medial aspect of the AON, in contrast to vehicle-treated animals (Fig. 1b).

The hippocampus represents spatiotemporal context and mediates memory retrieval by reinstating patterns of neocortical activity observed during learning[12]. Hippocampal projections to the main olfactory system are exclusive to the AON and are necessary for the recollection of contextually relevant odour information[26,27]. Indeed, context alone can drive the activation of specific odour representations in the absence of olfactory stimuli[29]. Repeated exposure to sensory stimuli decreases their salience. For odours, this habituation is behaviourally expressed as a decrease in investigative sniffing towards the stimulus[30]. If bottom-up experience sufficiently differs from top-down expectation, a mismatch is detected by the animal which dishabituates and attends to the stimulus[31]. This in turn can be measured as an increase in investigative sniffing. This paradigm was exploited to examine the necessity of AON neurons, active in response to an odour-context pairing, for mediating context-driven odour recollection.

Mice expressing hM4D-mCherry and reporter controls were repeatedly exposed to a cotton swab tip emitting a pure odourant for 30 min per day over 12 consecutive days (Fig. 1c). Odour exposure was conducted in a visually rich context, consisting of a 50 cm × 25 cm × 20 cm clear plexiglass cage with patterned wallpaper pasted on the exterior walls. Prior to the fifth exposure, the animals were injected with 4-OHT, allowing gene expression in active neurons. On day 13, the mice were administered clozapine-N-oxide (CNO; 5 mg/kg), a synthetic ligand for hM4D, and 15 min later, exposed to the training context in the absence of the odour. The mismatch between the contextual cue and the lack of an expected scent drives an increase in the amount of time spent investigating the cotton swab, as seen in mCherry-expressing controls (Fig. 1d)[27,29]. However, hM4D-mediated inhibition of tagged neurons resulted in the abolishment of this behaviour, suggesting memory for the odour was silenced. Importantly, CNO-injected control mice similarly trained but tested without the odour in a novel context displayed low levels of investigation, indicating context-specificity of the behavioural response.

We then evaluated the extent to which the tagged neurons were reactivated during retrieval of the odour memory (Fig. 1e, f). mCherry-expressing controls exhibited significantly greater reactivation of tagged neurons when exposed to the training context than a novel context (Fig. 1f). Evidently, reactivation in hM4D-expressing animals was suppressed, consistent with their poor behavioural expression of odour memory. Indeed, analysing a subset of mice from each group revealed a positive correlation between the reactivation rate and the time spent investigating the cotton swab (Fig. 1g). Importantly, the contextual cue alone was sufficient to produce elevated levels of c-Fos expression in the AON (Fig. 1f). Moreover, c-Fos expression did not differ between groups. This implies that hM4D-mediated inhibition of tagged neurons resulted in the emergence of a parallel network of active neurons which failed to support the odour memory. Similar to our observation, alternative populations of active neurons have previously been shown to appear following the artificial silencing of engrams in other brain areas[32]. Collectively, our results demonstrate that AON neurons involved in learning an

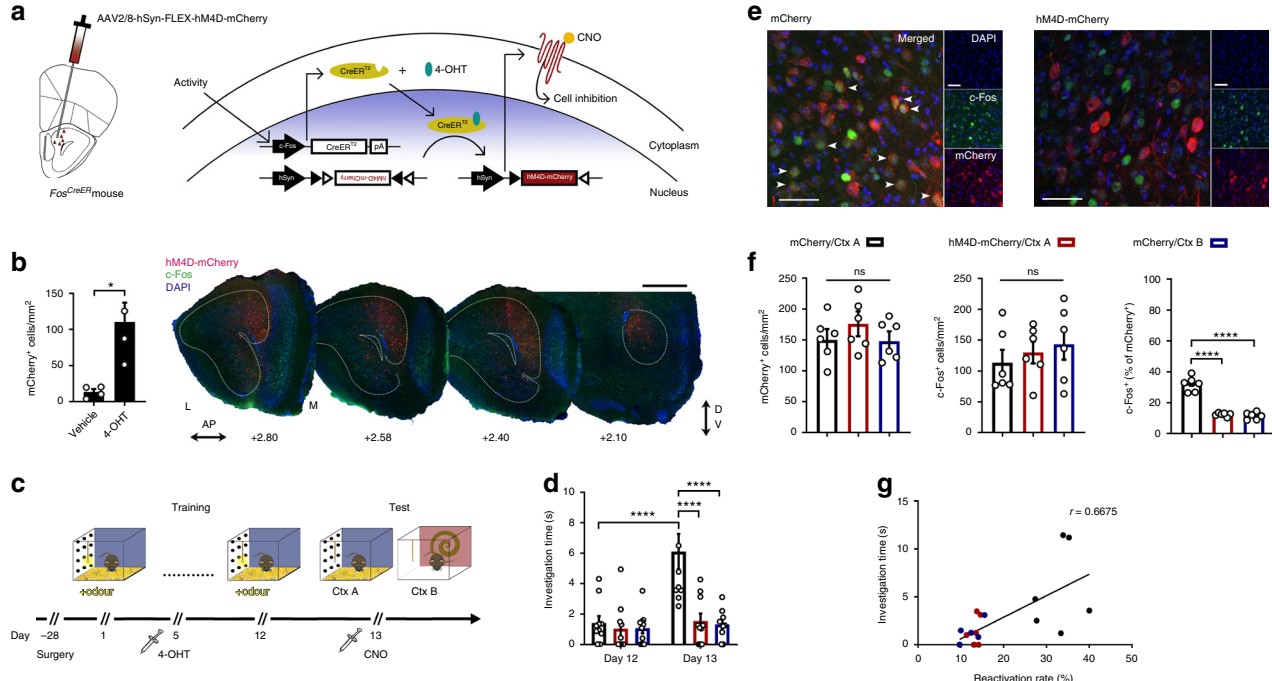

**Fig. 1 Silencing a context-specific odour memory. a** Method for activity-dependent labelling and cellular inhibition using hM4D-mCherry. **b** Injections of 4-OHT induces AAV-mediated gene expression restricted to the medial AON as seen in representative coronal sections of a $Fos^{CreER}$ mouse [each group, $n$ = 4 mice, two-tailed independent-samples $t$-test, $t(6)$ = 3.549, $^*P$ = 0.0121). Anterior/Posterior (AP); Dorsal/Ventral (DV). Coordinates are relative to bregma. Scale bar represents 500 μm. **c** Behavioural protocol for tagging and inhibiting context-driven odour engrams. Context (Ctx). **d** Behavioural expression of contextually-cued odour memory as measured by total investigation time of the odour source [each group, $n$ = 10 mice, two-way ANOVA, main effect of group $F(2, 54)$ = 11.44, $^{****}P$ < 0.0001; main effect of day $F(1, 54)$ = 13.38, $^{***}P$ = 0.0006; interaction between group and day $F(2, 54)$ = 8.421, $^{***}P$ = 0.0007]. **e** Context-driven reactivation of tagged AON neurons in representative confocal images. Scale bar represents 50 μm. Similar results were observed in all mice independently analysed. **f** Density measurements of labelled mCherry- (left) and c-Fos-positive (middle) neurons in the medial AON, and the corresponding reactivation rate (right) [each group, $n$ = 6 mice; mCherry densities: one-way ANOVA, $F(2, 15)$ = 0.7637, ns, $P$ = 0.4832; c-Fos densities: one-way ANOVA, $F(2, 15)$ = 0.492, ns, $P$ = 0.6209; engram reactivation: one-way ANOVA, $F(2, 15)$ = 81.43, $^{****}P$ < 0.0001]. **g** Correlation between the tagged AON neuron reactivation rate and investigation time [Pearson correlation coefficient, $^{**}P$ = 0.0025, $r$ = 0.6675]. Data are means ± SEM. Source data are provided as a Source Data file.

odour-context association are necessary for odour memory expression, supporting their identity as engrams.

**AON activity patterns are context-dependent.** Context can cue reinstatement of odour memory in the AON. However, it remains unclear whether the patterns of activity in the AON reflect the conjunctive odour-context experience or if it simply represents the olfactory component. To address this, AON neurons were tagged during exposure to an odour-context pairing in mice stereotaxically infused with the Cre-responsive mCherry reporter. One week after tagging, the animals were presented with the same odour either in the original context or a novel context. We found that animals placed in a novel context displayed significantly lower reactivation of the tagged population, compared to mice reintroduced to the original context (Supplementary Fig. 2). Despite presentation of the same odour, the distinctive responses in the AON indicate that the engram reflects context-specific, olfactory experiences.

**AON engram activity is sufficient for memory expression.** We next examined whether artificial activation of AON neurons is sufficient to elicit behavioural expression of odour memory. First, we probed the effects of hM3D-mediated activation of AON neurons under the context-driven odour recollection paradigm. Neurons active during an odour-context association were tagged with hM3D-mCherry or mCherry only (Supplementary

Fig. 3a–c). After training, the animals were injected with CNO and divided into two groups. One group was placed into the trained, familiar context and the other into a novel context, both lacking the associated odour. We found that hM3D-expressing animals investigated the cotton swab less than reporter controls within the familiar context, although the difference was not statistically significant (Supplementary Fig. 3d). In contrast, animals placed into a novel context exhibited greater investigation of the cotton swab relative to controls upon activation of the tagged neurons.

The activation data obtained thus far allows multiple interpretations owing to the inherent ambiguity of the 'investigation time' measure. To gain more direct behavioural evidence that specifically addresses the content of AON engrams, we designed a protocol that exploits the animal's goal-directed, foraging behaviour. For subsequent gain-of-function experiments, we used $Fos^{CreER}$ mice bilaterally infused with a Cre-dependent AAV expressing a channelrhodopsin variant, ChETA (AAV9-EF1α-DIO-ChETA-eYFP), or the reporter only (AAV5-EF1α-DIO-eYFP), with bilateral optic fibre implantations targeted above the AON (Fig. 2a, b). We confirmed that 4-OHT injections, followed by exposure to an odour-context pairing, resulted in ChETA-eYFP expression in AON neurons and that c-Fos expression within these cells increased upon blue light (473 nm) illumination (Fig. 2d, e).

Our behavioural paradigm was designed to delineate between two patterns of activity within the AON, each corresponding to a

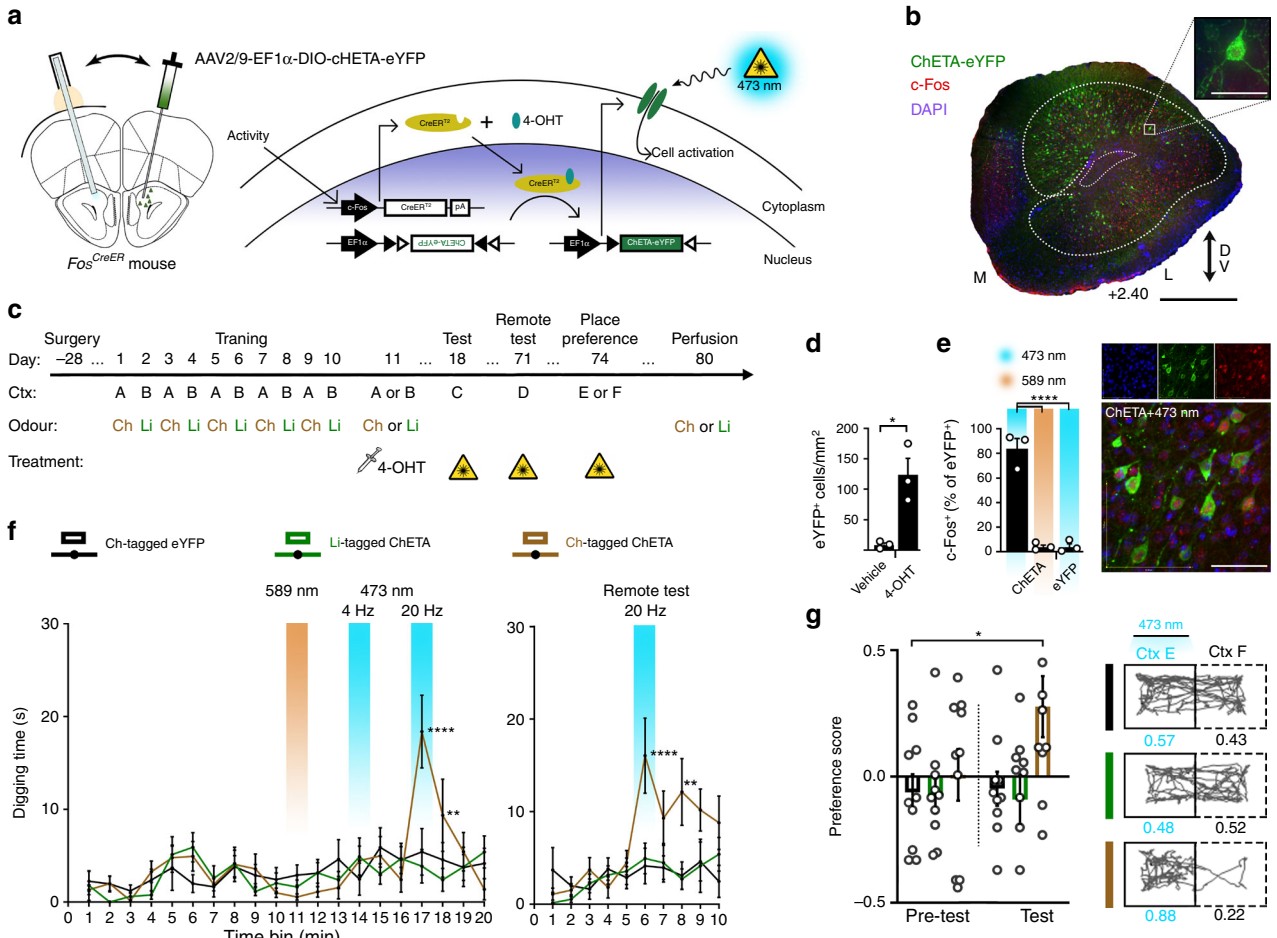

**Fig. 2 Artificially expressing unique odour engrams. a** Method for activity-dependent labelling of AON neurons with photoreceptive ChETA-eYFP for subsequent cellular activation. **b** Representative coronal section of the AON expressing ChETA-eYFP (green) and c-Fos (red); DAPI (blue). Scale bars represents 500 μm (black) or 50 μm (white). Medial (M); Lateral (L); Dorsal/Ventral (DV). Similar results were observed in all mice independently analysed. **c** Protocol for tagging and examining the behavioural expression of two unique odour engrams. Chocolate (Ch); Limonene (Li). **d** Injections of 4-OHT induces AAV-mediated gene expression of ChETA-eYFP [each group, $n = 3$ mice, independent-samples $t$-test, $t(4) = 4.184$, $*P = 0.0139$]. **e** Left: percentage of ChETA-eYFP or eYFP-only labelled neurons which express c-Fos following light illumination [each group, $n = 3$ mice, one-way ANOVA, $F(2, 6) = 79.67$, $****P < 0.0001$]. Right: representative confocal image of ChETA-eYFP neurons after stimulation with 473 nm light. Scale bar represents 50 μm. **f** Measurements of time spent digging 7 days (left) or 60 days (right) post-tagging [each group, $n = 10$ mice; 7 days (left): two-way ANOVA, main effect of group $F(2, 27) = 0.6025$, ns, $P = 0.5547$; main effect of Time $F(19, 513) = 4.901$, $****P < 0.0001$; main effect of subjects (matching) $F(27, 513) = 2.870$, $****P < 0.0001$; interaction $F(38, 513) = 2.352$, $****P < 0.0001$; 60 days: two-way ANOVA, main effect of group $F(2, 27) = 5.3$, $*P = 0.0114$; main effect of Time $F(9, 243) = 5.432$, $****P < 0.0001$; main effect of subjects (matching) $F(27, 243) = 2.929$, $****P < 0.0001$; interaction $F(18, 243) = 2.342$, $**P = 0.002$]. **g** Left: real-time place preference scores determined by the time spent in a 473-nm light-paired or unpaired chamber [each group, $n = 10$ mice, two-way ANOVA, main effect of group $F(2, 54) = 3.727$, $*P = 0.0305$; main effect of light $F(1, 54) = 1.492$, ns, $P = 0.2272$; interaction $F(2, 54) = 1.523$, ns, $P = 0.2272$]. Right: representative track plots of an individual mouse's position from eYFP- (top), limonene- (middle), or chocolate-tagged (bottom) groups. Data are means ± SEM. Source data are provided as a Source Data file.

unique odour engram that elicits a distinct behavioural output. Chocolate is a highly appetitive stimulus for mice and can act as a robust trigger of foraging behaviour, quantified as the amount of time spent digging. Moreover, mice can be conditioned to forage within a barren context previously associated with chocolate, implying memory for the chocolate odour alone can drive digging (Supplementary Fig. 4). Limonene, however, is a neutral odour that does not evoke an innate response in mice. We trained the animals for 10 days on alternating exposures of two odour-context associations- context 'A' in the presence of chocolate (Ctx A/Ch) or in a distinct context 'B' paired with limonene (Ctx B/Li) (Fig. 2c). On day 11, the mice were injected with 4-OHT and exposed to either the Ctx A/Ch or the Ctx B/Li condition. This protocol ensures that internal representations of both chocolate and limonene are formed, but the pattern of activity evoked by

each odour is tagged separately in designated groups of mice. One week after tagging, the animals were placed in a novel context 'C' in the absence of an applied odour and their digging behaviour scored (Fig. 2f). Expectedly, all groups showed no increase in the time spent digging in response to yellow light (589 nm) illumination. Notably, however, we found that high frequency (20 Hz) but not low frequency (4 Hz) stimulation was sufficient to elicit robust digging behaviour in ChETA-expressing mice tagged with the chocolate engram (Supplementary Movie 1). Incidentally, 20 Hz stimulation falls within the range of the beta band (15–30 Hz) coupling observed between the hippocampus and the olfactory bulb during odour association learning[33].

The lack of an explicit response in limonene-tagged mice further supports the existence of odour-specific engrams stored within the AON. Though, the possibility remains that additional

cells specifically coding for the rewarding aspects of chocolate were involved in the tagged population. As a control, we repeated our odour memory induction assay, yet AON neurons were tagged during presentation of the odour-associated contexts only. Under this condition, mice tagged in the chocolate-associated context still exhibited robust digging in response to optogenetic activation of AON neurons (Supplementary Fig. 5). Lastly, our model suggests that odour memories within the AON should form rapidly with experience, requiring only a single-trial learning event. To test whether the AON encodes episodic information, we tagged neurons with ChETA-eYFP or eYFP during the initial exposure to chocolate, and stimulated the tagged population after 7 days with no further interim training. Nevertheless, ChETA-expressing mice displayed a reliable digging response to blue light compared to controls (Supplementary Fig. 6).

**AON engrams are enduring.** To examine whether the tagged populations could support long-term odour memory retrieval, mice were exposed to a novel context 'D' in the absence of an applied odour 60 days post-tagging. Remarkably, chocolate-tagged, ChETA-expressing animals displayed an increase in the amount of time spent digging under 20 Hz blue light illumination, consistent with chocolate odour recollection (Fig. 2f; Supplementary Movie 2). To further probe the qualitative differences perceived during engram activation, we performed a real-time place preference test such that mice were allowed to explore two distinct, yet connected contexts 'E' and 'F', only one of which was paired with light. We found that chocolate-tagged mice readily developed a preference for the light-paired context compared to limonene- or eYFP-tagged controls which explored both contexts for relatively equal extents (Fig. 2g; Supplementary Table 1).

We also sought to determine the degree of overlap between the tagged population and the pattern of activity evoked by re-exposure to either the Ctx A/Ch or Ctx B/Li training conditions, 69 days post-tagging. Both chocolate- and limonene-tagged mice displayed similar levels of activation and eYFP-labelling densities within the AON (Fig. 3a, b). Chocolate-tagged mice exhibited greater reactivation following exposure to the Ctx A/Ch condition but not the Ctx B/Li, while limonene-tagged mice displayed the opposite pattern of activity (Fig. 3c, d). Thus, exposure to an odour-context pairing that is congruent to the condition

experienced at the time of tagging resulted in a higher reactivation of the original engram compared to incongruent configurations. These findings illustrate an enduring capacity for AON odour engrams to be expressed weeks after initial formation.

## Discussion

Odours are powerful stimuli used by most organisms to navigate their surroundings and find food, predators and mates[22]. Commonly attributed to its exclusive access to the limbic system, olfaction possesses a particularly evocative ability to generate and retrieve memories[27]. As a consequence, many attempts have been made to identify the neural substrate of odour memory[34–37]. Yet, to our knowledge no investigation has offered the cellular resolution analysis critical for dissecting the involved circuitry. Our results provide strong support for the AON as the storehouse of odour engrams (Fig. 4). We demonstrated that AON odour engrams are relatively enduring and that their activity is necessary and sufficient for the behavioural expression of odour memory. Our findings satisfy the persistence, dormancy, ecphory (i.e. capacity for cued recall), and content criteria, widely accepted to be crucial for engram identification[4]. Furthermore, the introduction of our chocolate foraging assay expands the behavioural study of memory to include a digging measure among the established freezing and approach/avoidance paradigms.

The AON likely contributes to mismatch detection between experience and expectation, a role supported by its anatomical position between bottom-up and top-down olfactory processing streams. The storage of odour engrams in the AON likely serves, by virtue of memory retrieval, many of the animal's odour-specific cognitive and behavioural demands. Yet, how the engram supports this wide range of olfactory functions is an open question that can be addressed in future studies. Importantly, the differential behavioural outcomes produced by AON activation carries the implication that the output is flexible, state-dependent and determined downstream the AON. The olfactory bulb and piriform cortex receive the bulk of the AON's projections, thus computations relating to odour memory expression are likely performed by these structures[24]. Consistently, piriform ensembles are known to elicit adaptive odour-guided behaviours, potentially acting as a functional link for the AON to motor pathways[38]. Ultimately, the phylogenetic conservation of the olfactory system

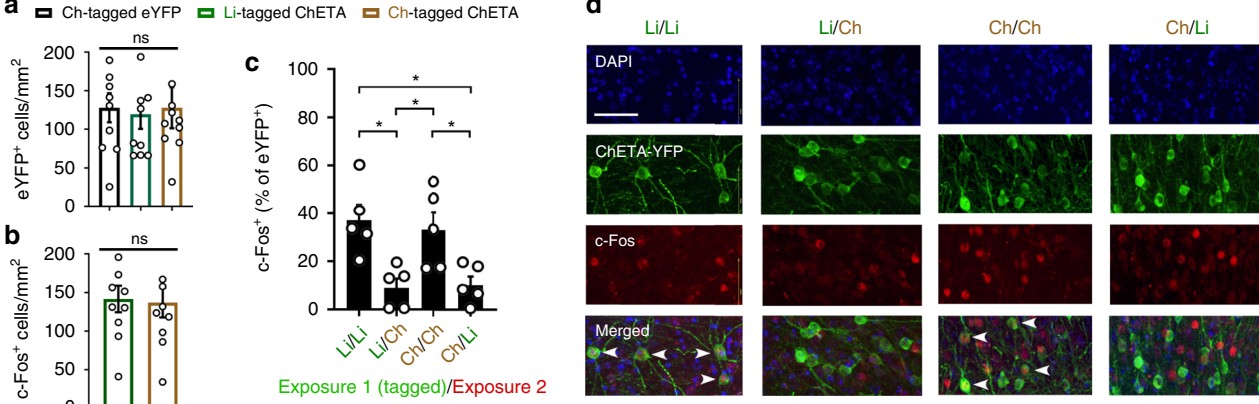

**Fig. 3 Long-term reactivation of odour engrams. a** Quantification of eYFP-labelling across groups [each group, $n = 10$ mice, one-way ANOVA, $F_{(2, 27)} = 0.04997$, ns, $P = 0.9513$]. **b** Medial AON activation as measured using c-Fos density as a proxy [each group, $n = 10$ mice, two-tailed independent-samples $t$-test, $t_{(18)} = 0.1807$, ns, $P = 0.8586$]. **c** Reactivation rate of limonene- or chocolate-tagged engrams following exposure to limonene or chocolate odours in their associated contexts, 69 days after initial tagging [each group, $n = 5$ mice, one-way ANOVA, $F_{(3, 16)} = 6.968$, **$P = 0.0033$, significance for multiple comparisons *$P < 0.05$]. **d** Representative confocal sections of each experimental condition. Scale bar represents 50 μm. Data are means ± SEM. Source data are provided as a Source Data file.

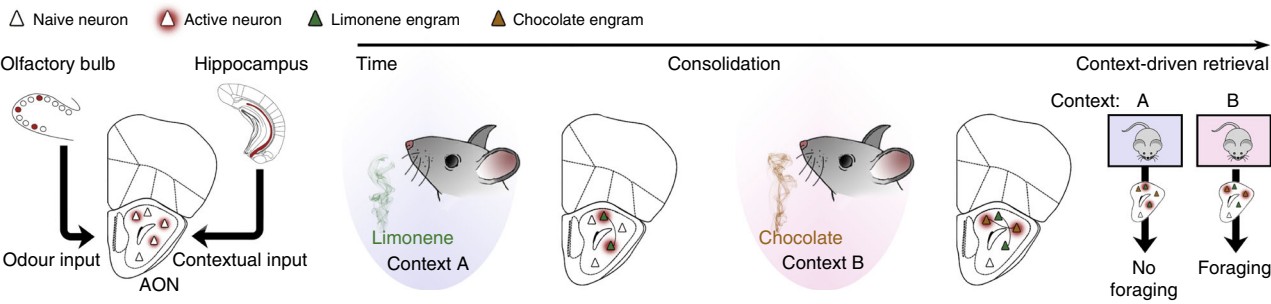

**Fig. 4 Odour engrams are stored in the AON.** A model diagram depicting the formation, storage and retrieval of unique, contextually-relevant odour engrams within the AON.

has afforded the AON and its homologous structures the ability to serve as an effective model for investigating memory and cognition across species.

## Methods

**Animals.** *Fos^CreER* mice obtained from the Jackson Laboratory and bred in-house at the Biological Sciences Facility were used for all experiments. Animals were group-housed in a temperature-controlled room on a 12-h light/dark cycle with ad libitum access to food and water. Surgery was performed on mice 8–10 weeks old, after which they were individually housed. Fifty-four mice were distributed into three groups for chemogenetic behavioural testing (hM4D-mCherry: $n = 10$, hM3D-mCherry: $n = 12$ mCherry controls: $n = 32$) and another 54 were used in optogenetic experiments (Ch-tagged eYFP: $n = 16$, Ch-tagged ChETA: $n = 16$, Li-tagged ChETA: $n = 10$, Ctx A-tagged ChETA: $n = 6$ and Ctx B-tagged ChETA: $n = 6$). Ten animals were used to examine context-driven digging behaviour. An additional 65 mice were used for histological purposes. All procedures were performed in accordance with the guidelines of the Canadian Council on Animal Care and the University of Toronto Animal Care Committee.

**AAV vectors.** AAV2/8-hSyn-FLEX-hM4D-mCherry and AAV2/9-EF1α-DIO-ChETA-eYFP were purchased from the University of Pennsylvania Vector Core and AAV2/8-hSyn-DIO-mCherry, AAV2/5-EF1α-DIO-eYFP and AAV2/9-CAG-FLEX-GFP from the vector core at the University of North Carolina at Chapel Hill.

**Surgical procedures.** Mice were anaesthetised with isoflurane (4% induction, 2% maintenance) and administered ketoprofen (5 mg/kg) for pain management. Viral vectors were bilaterally infused into the medial aspect of the AON (10° angle toward the midline, A/P: +3.00, M/L: ±1.15, D/V: −3.90) at a volume of 0.2–0.3 μL. Infusions were conducted using pressure ejection at a rate of 0.1 μL/min through a cannula connected by 20 cm of Tygon tubing to a 10-μL Hamilton syringe (Hamilton, Reno, NV). All infusions were allotted a 20-min interval to limit the viral spread. Optic fibres (200 μm core diameter, 0.39 NA; Thorlabs, Newton, NJ, USA) threaded through 1.25 mm-wide zirconia ferrules (Thorlabs) were implanted immediately dorsal to the infusion site (D/V: −3.80).

**Drugs.** Clozapine-N-oxide (CNO) was obtained from the NIH and dissolved in a solution consisting of 10% dimethyl sulphoxide and 0.9% saline. A 5-mg/kg dose was administered to all animals in chemogenetic behavioural tests. 4-hydroxytamoxifen (4-OHT, Sigma-Aldrich, Cat# H6278) was dissolved in 100% ethanol (20 mg/mL) and diluted using corn oil. The ethanol was allowed to evaporate, resulting in a final concentration of 10 mg/mL. All 4-OHT injections (50 mg/kg) were prepared fresh prior to intraperitoneal (IP) administration. To limit unintended labelling, all animals were transported in their homecage and placed into procedure rooms at least 2 h before 4-OHT injection and 2 h after testing.

**Optogenetic apparatus.** Activation of ChETA-eYFP-tagged neurons in the AON was achieved by pulsed illumination with blue light (473 nm, 2 mW, 10 ms pulse width). Stimulation was provided for a duration 1 minute at specific intervals indicated for each test. The stimulation frequency (4 Hz or 20 Hz) was produced using an arbitrary waveform generator (Agilent, 33220A). We delivered light generated with a diode-pumped solid-state laser (Laserglow, Toronto, Canada) through a 1 × 2 optical commutator (Doric Lenses, Quebec, QC, Canada) to divide the light path through two arena patch cables attached to the implanted optic fibres.

**Behavioural testing.** Our paradigms make use of two methods for creating different contexts. Transparent cages of the same size were used in each case with the exception of the place preference test. One method involved pasting visually patterned wallpaper on the exterior walls. Wallpapers depicting contrasting designs were chosen to represent different 'visual' contexts (e.g. black and white stripes vs.

colourful polka dots). Another method used transparent cages with different colourful objects (plastic toys, coffee cups, etc) placed around the outside of the cage. The real-time place preference test had the additional factor of metal fretwork flooring that differed between the two contexts. All designs/objects used were kept constant for each defined context. The apparatus was cleaned using 70% ethanol and wiped dry between trials.

Context-driven odour recollection: this paradigm was adapted from Mandairon et al. (2014) and previously described by the authors[27]. Mice were conditioned to an odour within a visually rich context consisting of a 50 cm × 25 cm × 20 cm clear Plexiglas cage with patterned designs pasted on the exterior of the walls. A cotton-tipped wooden applicator was attached to the centre of the cage lid such that it was positioned ~3 cm from the floor bedding. Before introducing mice into the chamber, 100 μL of a monomolecular neutral odourant (pseudorandomized limonene, isoamyl acetate, or 1-pentanol) was applied to the cotton tip. The mice were allowed 1 h to explore the odour-imbued cotton swab within the context over 12 consecutive days. On day 5, the animals were placed in the behavioural testing room 2 h before injection with 4-OHT to reduce unintended labelling. Ten minutes after treatment with 4-OHT, the animals were exposed to the context-odour pairing. The day after conditioning and 15 min prior to testing, the mice were injected with CNO and placed into the context in the absence of the applied odour. A subset of the mCherry-control mice were instead put into a previously unexplored context. Behaviour during exploration on Day 12 and 13 was filmed during the initial 5 min. A NIKON D5200 equipped with a 35-mm f/1.8G lens was used to record at 60 fps. Using a reduced frame rate, the amount of time spent investigating the cotton tip was subsequently scored blind to the experimental group. Investigations were strictly defined as head up sniffing, directed towards and within 1 cm of the cotton swab tip.

Contextually-cued digging: animals were placed in a clear 50 cm × 25 cm × 20 cm Plexiglas cage surrounded by colourful objects. The cages contained ~15 g of chocolate spread (Nutella, Ferrero, Italy) placed in a 3-cm wide × 1 cm high aluminium cup covered underneath the bedding. Each mouse remained in the context for 30 min or until the chocolate was uncovered. Training repeated daily for 5 consecutive days. The following day, animals were placed back into the original context or into a novel context. Behaviour was filmed for 5 min and the amount of time spent digging was subsequently scored. Digging was defined as the duration elapsed with the nose pointed downwards and forepaws engaged in uncovering bedding. All animals were food deprived overnight before testing.

Odour memory induction assay: animals were exposed to alternating 1-h presentations of two distinct odour-context configurations over 10 consecutive days. Both contexts contained a cotton-tipped wooden applicator positioned 3 cm from the floor and 5 cm from one end of the chamber They also contained an aluminium cup hidden within the bedding 5 cm from the opposite side. The Ctx A/Ch condition involved placing the animals in a 50 cm × 25 cm ×20 cm clear Plexiglas cage surrounded by colourful objects. Within this context, approximately 15 g of chocolate was added to the aluminium cup. The Ctx B/Li condition was composed of a similar Plexiglas cage with striped and dotted visual patterns pasted onto the outer walls. No chocolate was added to the aluminium cup, instead 100 μL of limonene was applied to the cotton-tip. On day 11, animals were divided into three groups and injected with 4-OHT. Ten minutes after injection, two groups were exposed to the Ctx A/Ch configuration (Ch-tagged ChETA or eYFP) while the third group was instead presented with the Ctx B/Li combination (Li-tagged ChETA). Seven days later, the animals were placed into a novel context 'C' and their behaviour filmed for 20 min. On the 11th minute animals were illuminated with constant yellow light (589 nm). During the 14th and 17th minute, the animals were stimulated using blue light and a frequency of 4 Hz or 20 Hz, respectively. All animals were food deprived overnight before testing. During the remote test 53 days later, animals were reintroduced to context C for 10 min and stimulated with 20 Hz blue light during the 6th minute. All videos were subsequently scored blind to the experimental group for the amount of time spent digging.

Real-time place preference: a 45 cm × 20 cm × 35 cm apparatus was equally divided such that each side possessed a distinct wallpaper and metal fretwork flooring. One side, pseudorandomized, was paired with 20 Hz blue light illumination when animals entered its boundaries. The mice were first allowed to explore both chambers

with the laser off for a 10-min habituation period. Animals were then returned to their homecage for a 10-min inter-trial interval. Mice were reintroduced to the context with the laser on for 5 min. Overhead videos were obtained using a Logitech webcam and analysed using ANY-maze software (Stoelting Co) to determine the amount of time animals spent in each chamber and their corresponding track plots.

**General histology**. Mice were transcardially perfused with 0.1% phosphate-buffered saline (PBS, pH 7.4), followed by 4% paraformaldehyde (PFA). The brain was then extracted and post-fixed in 4% PFA overnight. Next, the tissue was stored in a 30% sucrose solution at 4 °C for 72 h. Coronal sections (40 μm) were obtained using a cryostat (Leica, Germany), slide-mounted and counterstained with DAPI (4′,6-diamidino-2-phenylindole) for 5 min before adding a coverslip with Aqua-mount (Polysciences, Inc., Warrington, PA). Wide-field fluorescent images were captured using a ×4 objective lens on a fluorescent microscope (Olympus, Japan). Confocal images were captured using a ×20 and ×60 objective through a Quorum spinning disk confocal microscope (Zeiss, Germany). Adobe Photoshop CS6 (Adobe Systems, Incorporated, San Jose, CA) was used to adjust the brightness and contrast of representative sections.

**c-Fos immunostaining**. Mice were perfused and their brains extracted 75 min after initial exposure to the experimental condition. Free-floating tissue sections were collected, washed with PBS containing Triton X-100 (PBST), and blocked with 5% normal donkey serum for one hour. The tissue was later incubated at 4 °C for 72 h with rabbit polyclonal anti-c-Fos antibody (1:1000 in PBST; Santa Cruz Bio-technology, California). Next, the sections were incubated at room temperature with Alexa Flour 594-conjugated (or Alexa Flour 488-conjugated) donkey anti-rabbit secondary antibody (1:500 in PBST; Invitrogen) for 2 h.

**Cell counting**. Cell counting was performed using ImageJ software (NIH). For every animal, the medial aspect of the AON was outlined in every section, forming multiple regions of interest (ROI). The number of labelled cells in each ROI was quantified and divided by the area to obtain a density measure. The mean density was determined for each animal and each experimental group accordingly. Parameters were calibrated for counting using randomly chosen representative neurons within the AON and kept constant for each animal. Signal detection and circularity thresholds were adjusted to provide an accurate, yet conservative estimate.

**Calculations and statistical analysis**. The reactivation rate was calculated according the following formulas:

hM4D-mCherry experiments:

$$\text{Reactivation rate} = \frac{\sum (\text{cFos}^+ \cdot \text{mCherry}^+ \text{ cells})}{\sum (\text{mCherry}^+ \text{ cells})} \times 100\%$$

ChETA-eYFP experiments:

$$\text{Reactivation rate} = \frac{\sum (\text{cFos}^+ \cdot \text{eYFP}^+ \text{ cells})}{\sum (\text{eYFP}^+ \text{ cells})} \times 100\%$$

Sample sizes were chosen on the basis of previous studies. GraphPad Prism version 7.04 for Windows (GraphPad Software, La Jolla, CA, USA) was used for performing statistical analyses. Significance was determined using a (two-tailed) unpaired Student's $t$-test, (two-tailed) one-sample $t$-test, ordinary one-way analysis of variance (ANOVA), two-way repeated-measures ANOVA, and Pearson correlation where appropriate. Tukey's multiple comparisons test was used for post hoc comparisons. Significance was defined as $*P < 0.05$, $**P < 0.005$, $***P < 0.0005$, $****P < 0.0001$.

**Reporting summary**. Further information on research design is available in the Nature Research Reporting Summary linked to this article.

## Data availability
All relevant data are available from the corresponding authors upon reasonable request. The source data underlying Fig. 1b, d, f, 2d–g and 3a–c and Supplementary Figs. 1a, b, 2, 3b, d, 4b, 5 and 6 are provided as a Source Data file.

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

## Acknowledgements

This research was funded by operating grants to J.C.K. from the Canadian Institutes of Health Research (CIHR) (MOP 496401) and the Natural Sciences and Engineering Council of Canada (NSERC) (MOP 491009). A.J.A. was supported by the NSERC postgraduate scholarships-Doctoral Program (Appl ID: PGSD2- 519452 - 2018).

## Author contributions

A.J.A. and J.C.K. carried out the study conceptualisation and experimental design. A.J.A. conducted the investigation and formal analysis. A.J.A. and J.C.K. wrote the manuscript.

## Competing interests

The authors declare no competing interests.
