## [Peer Review File · Nature Communications]

Reviewers' Comments:

Reviewer #1:

Remarks to the Author:

Agrabawi and Kim investigate context-odor memories in the anterior olfactory nucleus (AON), focusing on cFos+ cells and their genetic tagging and manipulations ("engram experiments"). In previous published work they had shown that activity in AON is important for odor perception, that ventral hippocampus projects to AON, and that this projection is important for coincidence detection (context-odor) and recollection of odor memory. Here they show that cFos+ cells are only detected in AOL upon exposure of mice to novel context AND odor (confirming previous findings at the level of cFos+ cells). They then show that: 1) activity in such cFos+ AOL cells is necessary for mice to look for missing odor in a context that had been associated with a particular odor; 2) activation of chocolate-context cFos+ cells in AOL is sufficient to induce digging (whereas those associated with limonene are not); and that 3) these memory traces are long-lasting. They conclude that the cFos+ neurons in AOL fulfil the criteria for representing an engram of odour-context memories.

The experiments in this study are carried out at a good quality level, and the outcomes are convincing. The case for a valuable paradigm to study memories under well-defined conditions is adequately made. On the other hand, none of the findings are particularly novel when taken in isolation: the link between AOL and behaviourally important context-odor memories had been made before, and properties of "engram cells" have been documented in many systems (including hippocampus and amygdala) by several previous studies. As a consequence, the advance provided by this study is limited.

In the opinion of this reviewer, the study could be strengthened by two additional experiments:

1) the loss- and gain of function experiments (Figs, 1 and 2) do not involve the same behavioural protocol. I would recommend adding an experiment to Fig1 in which one would activate the cFos+ cells (instead of inhibiting them) in Context-B with and without odor: will mice now look for the odor? only when it is missing?

2) odour-context memory should form very rapidly and not require 10-15 days of (over)training. I wonder whether AOL has a role in detecting mismatch from robust expectations, as opposed to just storing odor-context memories for retrieval. I would therefore suggest repeating the chocolate/limonene experiments upon only 1-2days of training, followed by tagging, waiting for sufficient time for viral expression, and then testing.

Minor point: I don't think it is accurate to state that in Fig1 memories are "erased" - one of their important counterparts was silenced.

Reviewer #2:

Remarks to the Author:

This study uses activity-dependent labelling in combination with optogenetic and chemogenetic tools to examine olfactory memory representations in the anterior olfactory nucleus (AON) in awake mice. The authors found that odor specific engrams are stored in the AON and that these engrams are relevant for behavioral expression of odor memory. Overall, this is an interesting study that examines an under investigated system in the olfactory pathway. I do have, however a number of concerns:

Major Concerns

1) The results of the activity dependent labelling are hard to understand: The authors report that that neither a novel odor or a novel context ALONE increased the number of labelled AON cells.

- how can this be interpreted in the light that clear odor responses (without changing context) as well as spontaneous sniff coupled activity could be observed in multiple studies (Lei et al., 2006; Kikuta et al., 2010; Rothermel and Wachowiak, 2014). These studies suggest that even in the absence of a sensory stimulus (due to changes in sniff rate) or at least in the odor condition, activity dependent labelling in the AON should be observable.

- the example pictures in Supp. Fig 1 are far too small to judge if the activity dependent labelling approach is successful. The whole AON should be shown for the different conditions instead of the magnification that depict few cells in the medial AON.

- the authors should additionally show a positive control e.g. the piriform cortex where clear odor responses (labelling) irrespective of context should be observable.

- why was an activity dependent labelling possible at the fifth exposure to a spatial context and odor (line 60)? Before, the authors claim that a novel odor-context pairing is necessary for including AON activity. A fifth presentation clearly is not new anymore, so no labelling should be observable?! The same paradigm was even used for 12 consecutive days (line 200). Alternatively, if the context does not have to be new, how was an odorant applied without any context so that no activity depending labelling was observed?

- Line 77 "Importantly, the contextual cue alone was sufficient to produce elevated levels of c-Fos expression in the AON (Fig. 1f)". Also this is contradictory to what the authors claim at the beginning.

- which neurons are mainly tagged by the activity dependent labelling? Excitatory projections neurons? Local inhibitory neurons?

- line 200. The mouse line used does not mark cells activated FOLLOWING 4-OHT injection but all cells that show a CRE Expression at the time of the injections. Taken protein turnover rate into account this can label neurons that were activated many hours before 4-OHT injection. Appropriate controls should be included.

2) Line66 "Importantly, control mice similarly trained but tested in a novel context displayed low levels of investigation, indicating context-specificity of the behavioural response". How can this be? This should be the "baseline" condition and a novel context should induce HIGH levels of investigation...

3) Statistical tests (which one, p values etc.) should be clearly stated in the main text!

4) Line 78 "Moreover, c-Fos expression did not differ between groups. This implies that hM4D79 mediated inhibition of tagged neurons resulted in the emergence of a parallel network of active neurons which failed to support the odour memory." This is strange. Why should now a completely different set of cells be active?

5) Line 101 "This protocol ensures that internal representations of both chocolate and limonene are formed, but the pattern of activity produced by each odour is tagged separately in designated groups of mice." Are the amount of labelled cells the same for the two engrams?

Even if not, additional cells might be tagged that specifically code for the reward and not necessarily for the odor component. Control experiments should be included.

Other Points

- What type of control were used in the study? Reporter only injected animal (line 56) or vehicle-treated animals (line 58)

- the optical stimulation protocol should be better described. E.g. what about the stim strength; was it homogenous between animal? At what stimulation strength was an effect observed. Was the 4 Hz

protocol tested with higher stim strength? Was there any correction for the duty cycle etc...

- "ecphory" should be better explained (Line 142)

- how high was the dropout rate for the 4-OHT injected animals

- the injection plane (Bregma +3.0) is close to the OB and other olfactory structures. Retrograde infection and the possibility of antidromic stimulation should be evaluated (e.g. by showing histological sections from the OB, piriform, hippocampus).

- Line 229: why is there no control for the third experimental group?

References

Kikuta S, Sato K, Kashiwadani H, Tsunoda K, Yamasoba T, Mori K (2010) Neurons in the anterior olfactory nucleus pars externa detect right or left localization of odor sources. *Proc Natl Acad Sci U S A* 107:12363-12368.

Lei H, Mooney R, Katz LC (2006) Synaptic Integration of Olfactory Information in Mouse Anterior Olfactory Nucleus. *J Neurosci* 26:12023-12032.

Rothermel M, Wachowiak M (2014) Functional imaging of cortical feedback projections to the olfactory bulb. *Front Neural Circuits* 8:73.

Reviewer #3:

Remarks to the Author:

In this manuscript, Aqrabawi and Kim address the intriguing question: does the anterior olfactory nucleus (AON) contain odor-specific engrams. Specifically, they address whether the experience of encountering an odor in a specific environment (i.e., conjunctive experience of specific odor + specific context) is represented in AON. They address this by silencing, as well as optogenetically activating, medial portions of AON neurons that are tagged in an activity-dependent (c-Fos-driven) manner, with the aim to test if AON is necessary and sufficient for reactivating context-dependent olfactory experiences. They do so using two behavioral paradigms: (1) the investigative behavior when an odor is removed from a familiar context, and (2) comparison of food-seeking behavior in the presence of chocolate odor in one context vs. in the presence of less appetitive odor in a second context. They use the paradigm (1) for testing the necessity and paradigm (2) for testing the sufficiency.

The question is interesting and an important one. In general, the technique of tagging neurons using the c-Fos pathway seems to work, as many others have shown before for other brain regions. The major concern I have regarding this work is the design and rationales of the behavioral experiments. As explained below, the paradigms used cannot directly distinguish whether the tagged neurons participate in the associative memory (odor + context) or just the olfactory aspects. It is made even more difficult by the fact that rationales for the experimental designs are not clearly explained. As such, in my opinion, with various alternative hypotheses that remain, the data presented currently are, unfortunately, weak and do not necessarily support the claim that AON neurons represent olfactory experiences in specific contexts.

Major concerns:

While the authors claim that the tagged neurons represent associative experience of odor + context, it is difficult to distinguish if the neurons represent odor only or odor + context. This is especially the case for the experiment presented in Figure 2. In this dataset, optogenetic activation of tagged

neurons leads to food-seeking behavior only when the tagging occurred while mice were exposed to chocolate odor in a specific environment. However, it is also very likely that the chocolate odor on its own would induce this behavior once the conjunction [chocolate odor -> chocolate availability] has been learned. That is, the data could easily be explained as the tagged AON neurons representing the olfactory stimuli, which could then drive food-seeking behavior downstream. A more convincing experiment would be to tag AON neurons when only the context "A" is presented. The supplementary figure 2 shows that mere exposure to the context "A" leads to reactivation of the memory and thus food-seeking behavior. If tagging AON neurons in the absence of odor this way leads to food-seeking behavior, it would dissociate the two alternative hypotheses (AON representing odor only vs. odor + context) and be a convincing result indeed.

As for the first behavioral paradigm presented in Fig 1, this seems to be assessing novelty detection. The rationale for using this paradigm, in particular, an explanation on what is driving the mice to investigate the cotton swab, is not given clearly. Is it the absence of an odor when it is expected, or is it that the swab on its own smells different, and as a result regarded as a novel object? In addition, if the tagged AON neurons indeed represented the conjunctive odor + context experience, does not silencing the tagged AON neurons lead to an entirely new experience when the mice are placed in the otherwise familiar context? How do mice behave overall? The manuscript presents only the time spent investigating the cotton swabs. As a result, it is difficult to understand what is going on. A crucial control that is missing here, in order to show an importance of context in the reactivated AON neurons, is an assessment of the overlap in cFos-expressing population and the mCherry-expressing population when mice are exposed to context "B" AND the familiar odor. This needs to be contrasted to the proportion of overlap when the "full" stimulus is given (context "A" + familiar odor). These data would give an estimate on the extent to which re-activation depends on odor only vs. odor + context.

The authors used two different behaviors to test the necessity and sufficiency – that is, the necessity for one behavior, and sufficiency for a different behavior. The first behavior assesses novelty detection, and the second, food-seeking behaviors. It is unclear why the authors did this.

Regarding the contexts: are the contextual cues represented in the AON visual, or olfactory? In order to distinguish the two, more clean way to change the visual context would be to create different contexts "virtually" – i.e., using the same cage and contents, but by projecting different images. If this is too difficult, the authors should at least detail how the cages had been cleaned and describe in more detail what parameters should differ between the two contexts.

Minor comments

Choi et al in 2011 (DOI 10.1016/j.cell.2011.07.041) previously showed that activity-dependent tagging of piriform neurons can drive opposing behaviors depending on the experience. Given that AON and the piriform cortex have dense interconnections, what distinct roles do authors think that the two regions have? In addition, the authors need to cite the above paper.

Regarding the data presented in Fig. 1f, the authors state that the data come from "a subset of mice" (line 75). Can the authors please elaborate?

In general, please describe the experiments more clearly and with more detail. For example, in Fig 1, when the mice are exposed to context B, was the familiar odor present? Were control mice given CNO (line 66)?

Line 59: Please explain the enrichment.

Line 62: Please explain how much later (e.g. hours or days).

Line 66: Can it be that the memory is made inaccessible, instead of erased?

Line 105: Please elaborate on the stimulation parameters, in particular stimulation duration.

Reviewer #1

Aqrabawi and Kim investigate context-odor memories in the anterior olfactory nucleus (AON), focusing on cFos+ cells and their genetic tagging and manipulations ("engram experiments"). In previous published work they had shown that activity in AON is important for odor perception, that ventral hippocampus projects to AON, and that this projection is important for coincidence detection (context-odor) and recollection of odor memory. Here they show that cFos+ cells are only detected in AOL upon exposure of mice to novel context AND odor (confirming previous findings at the level of cFos+ cells). They then show that: 1) activity in such cFos+ AOL cells is necessary for mice to look for missing odor in a context that had been associated with a particular odor; 2) activation of chocolate-context cFos+ cells in AOL is sufficient to induce digging (whereas those associated with limonene are not); and that 3) these memory traces are long-lasting. They conclude that the cFos+ neurons in AOL fulfil the criteria for representing an engram of odour-context memories.

The experiments in this study are carried out at a good quality level, and the outcomes are convincing. The case for a valuable paradigm to study memories under well-defined conditions is adequately made. On the other hand, none of the findings are particularly novel when taken in isolation: the link between AOL and behaviourally important context-odor memories had been made before, and properties of "engram cells" have been documented in many systems (including hippocampus and amygdala) by several previous studies. As a consequence, the advance provided by this study is limited.

In the opinion of this reviewer, the study could be strengthened by two additional experiments:

1) The loss- and gain of function experiments (Figs, 1 and 2) do not involve the same behavioural protocol. I would recommend adding an experiment to Fig1 in which one would activate the cFos+ cells (instead of inhibiting them) in Context-B with and without odor: will mice now look for the odor? only when it is missing?

We greatly appreciate the reviewer's insightful suggestion. The updated manuscript now includes a new Supplementary Fig. 3 which summarizes the results obtained from the context-driven odour recollection test (described in Fig. 1) during hM3D-mediated activation of tagged AON neurons. We found that hM3D-expressing mice investigated the cotton swab less than reporter controls in the familiar training context A, although the difference was not found to be statistically significant. In contrast, as the reviewer correctly predicted, hM3D mice showed greater investigation when presented with the cotton swab in a novel context B. This differential behavioural outcome carries important implications. It suggests that the behavioural output (i.e. cotton swab investigation) produced in response to the odour engram activation is flexible, likely mediated by downstream targets of the AON, and dependent on the animal's state. When the odour is perceived during the engram activation in the familiar context A, it would result in lower levels of mismatch detection, which is behaviourally displayed as low levels of investigation (similar to what is observed on Day 12). On the other hand, the odor engram activation in a novel context-B would constitute a novel context-odor pairing, therefore increasing levels of mismatch detection. Nonetheless, we fully acknowledge the inherent ambiguity of the behavioural measure used in this paradigm which makes interpretation challenging. Avoiding the potential ambiguity and difficult interpretation was our motivation to design the behavioural protocol used in our gain-of-function

experiments (Fig. 2) by exploiting the animal's goal-directed digging behaviour. We have made these points clear and expanded our reasoning in the updated manuscript.

2) Odour-context memory should form very rapidly and not require 10-15 days of (over)training. I wonder whether AON has a role in detecting mismatch from robust expectations, as opposed to just storing odour-context memories for retrieval. I would therefore suggest repeating the chocolate/limonene experiments upon only 1-2 days of training, followed by tagging, waiting for sufficient time for viral expression, and then testing.

While engrams have been shown to be dynamic (see for review Davis & Reijmers, 2018), it is unlikely that the population of neurons tagged in our protocols will differ significantly from those responding to the initial exposure, to the extent that they produce different behavioural outcomes when activated. Consistently, our c-Fos data (Fig. 1) demonstrates that a relatively high degree of overlap is observed between the neuronal populations active at the time of tagging (on day 5) and those during retrieval at distant time points. Indeed, this overlap is comparable to that seen in animals tagged upon initial exposure (Supplementary Fig. 2). In addition, the strengthened synapses resulting from the routine repetition of the same odour-context pairing has been shown to allow a more reliable neural representation to form, unique to the odour associated with the context (Mandairon et al., 2014; Shakhawat et al., 2014). Lastly, the measures of tagged neurons reported in this manuscript are comparable between animals tagged following a single exposure (Supplementary Fig. 1, 2; Fig. 1b; Fig. 2d) and those that were tagged after five days of training (Fig. 1f; Fig. 3a). While this particular comparison does not directly support our argument, it suggests at least that overall activity levels within the AON are maintained and do not differ across training sessions. Therefore, we believe repeating the chocolate/limonene experiments but tagging upon the initial exposure is unlikely to yield meaningful results. We were also discouraged to inject animals immediately prior to the initial exposure because the mice may develop an association between the aversive experience of the drug injection and the presented odour. Such an association would influence the animal's innate olfaction-dependent behaviour, and in turn the interpretation of our results.

Please note, however, that we agree with the reviewer; it is possible that the AON contributes to the detection of a mismatch between experience and expectation, credible due to its anatomical position between bottom-up and top-down olfactory processing streams. The storage of odour-context engrams in the AON likely serves, by virtue of subsequent memory retrieval, many of the animal's odour-specific cognitive and behavioural demands. How the engram is accessed and used to support a wide range of functions is an open question that can be addressed in future studies, yet is beyond the scope of the current work.

Other minor point:

I don't think it is accurate to state that in Fig1 memories are "erased" - one of their important counterparts was silenced.

The word 'erased' has now been replaced with 'silenced' in the updated manuscript to more accurately describe our interpretation of the data.

Reviewer #2

This study uses activity-dependent labelling in combination with optogenetic and chemogenetic tools to examine olfactory memory representations in the anterior olfactory nucleus (AON) in awake mice. The authors found that odor specific engrams are stored in the AON and that these engrams are relevant for behavioral expression of odor memory. Overall, this is an interesting study that examines an under investigated system in the olfactory pathway. I do have, however a number of concerns.

Major Concerns:

1) The results of the activity dependent labelling are hard to understand: The authors report that that neither a novel odor or a novel context ALONE increased the number of labelled AON cells.

- How can this be interpreted in the light that clear odor responses (without changing context) as well as spontaneous sniff coupled activity could be observed in multiple studies (Lei et al., 2006; Kikuta et al., 2010; Rothermel and Wachowiak, 2014). These studies suggest that even in the absence of a sensory stimulus (due to changes in sniff rate) or at least in the odor condition, activity dependent labelling in the AON should be observable.

We greatly appreciate the reviewer's comments, which helped us further clarify the experimental approach and data interpretation in the revised manuscript.

Please note that expression of c-Fos in AON pyramidal cells was observed in all mice, regardless of the experimental conditions. However, the number of labelled cells is lower in the homecage, odour-only, and context-only conditions (i.e., baseline levels) compared to the coincident-input condition (odour + context).

In our hands, exposure to the odour only (still in homecage) or to a novel context in the absence of any salient odours did not change the levels of c-Fos in the AON. The lack of c-Fos induction following the 'odour-only' condition is consistent with an earlier study by Kay, Meyer, Illig, and Brunjes (2011) looking into the spatial distribution of neural activity in the AON, evoked by odour and electrical stimulation. Our data shows that c-Fos expression can be significantly increased beyond baseline levels when animals are concurrently presented with an odour in a distinct context. These findings demonstrate that baseline activity, generated by sniffing in the homecage, can be augmented by converging and salient olfactory and contextual inputs to the AON. Since neither input alone was sufficient to increase AON activity beyond baseline, the AON must contain neurons that increase firing in response to coincident odour-context inputs.

The studies referred to by the reviewer indeed seem to differ from our observations on the changes in AON activity. However, the methodological differences (electrophysiological recording vs. c-fos immunohistochemistry) likely contribute to the discrepancy. Measuring changes in c-Fos immunoreactivity is the most commonly used, established way to validate the effect of chemo- and optogenetic activation, and it has been utilized with great success to map out neuronal pathways activated by specific stimuli (ex. Vetere et al., 2019). However, it may not exactly correlate with neuronal firing rates measured in various in vivo electrophysiological recordings. For example, a recent study showed

that c-Fos-expressing neurons are only a subset of neurons that fire action potentials during open field exploration (Tanaka et al., 2018). Therefore, our c-Fos counting data likely underestimates the number of neurons that are activated by different behavioural conditions.

In addition, the AON is a heterogeneous structure with functional specialization among its different subregions. Our study has focused on the medial aspect of the pars principalis, whereas Kikuta et al. (2010) examined the pars externa, known to have a fundamentally different cytoarchitecture, connectivity, and physiology. Furthermore, the works cited were conducted in a recording context outside the homecage which may constitute a context sufficiently different to induce activity when paired with an odour. Indeed, Rothermel and Wachowiak's (2014) study highlights the dependence of AON activity on salience since they found the strength of AON feedback signals to be greater during wakefulness.

- The example pictures in Supp. Fig 1 are far too small to judge if the activity dependent labelling approach is successful. The whole AON should be shown for the different conditions instead of the magnification that depict few cells in the medial??? AON.

Supplementary Fig. 1 now presents images of the whole AON.

- The authors should additionally show a positive control e.g. the piriform cortex where clear odor responses (labelling) irrespective of context should be observable.

The activity-dependent labelling is a result of the Cre-mediated expression of the gene virally delivered to neurons. Since the medial AON was targeted in our stereotaxic surgery, virtually no AAV-mediated delivery of the gene was made to the piriform cortex, therefore in our experiments the piriform cortex does not show expression under any condition.

- Why was an activity dependent labelling possible at the fifth exposure to a spatial context and odor (line 60)? Before, the authors claim that a novel odor-context pairing is necessary for including AON activity. A fifth presentation clearly is not new anymore, so no labelling should be observable?! The same paradigm was even used for 12 consecutive days (line 200). Alternatively, if the context does not have to be new, how was an odorant applied without any context so that no activity depending labelling was observed?

Novelty is not required for activity in the AON. Odour-context coincidence detection and mismatch with previous experience is sufficient to increase activity in the AON, as shown previously (Agrabawi and Kim, 2018) and in this manuscript, regardless of whether the coincident input is novel or familiar.

A salient 'context', in our view, refers to all spatiotemporal configurations outside the homecage. Thus the odour-only presentations were made in the animals homecage. The manuscript has been updated to improve clarity.

- Line 77 "Importantly, the contextual cue alone was sufficient to produce elevated levels of c-Fos expression in the AON (Fig. 1f)". Also this is contradictory to what the authors claim at the beginning.

While the AON does not respond to context-only stimuli, if a salient odour is concurrently paired with a distinct context, coincidence-detecting AON neurons are activated and the contextual cue alone can later induce AON activity in the absence of the odour stimulus (Mandairon et al., 2014). Importantly, optogenetic inhibition of hippocampal terminals carrying contextual information during retrieval selectively reduces this AON activity (Supplementary Fig. 5 in Agrabawi and Kim, 2018). These observations do not contradict but support the argument made in this study, that the AON stores odour memory representations. This is now clarified in the updated manuscript.

- Which neurons are mainly tagged by the activity dependent labelling? Excitatory projections neurons? Local inhibitory neurons?

Our approach did not involve cell type-specific promoters to mediate gene expression. Also, injections with 4-OHT opens an approximately 5-hour-long window for activity-dependent labelling. Thus, it is almost certain that both neuronal cell types were tagged. Nonetheless, glutamatergic pyramidal cells compose the vast majority of neurons within the ring of AON pars principalis, among sparse GABAergic cells present uniformly throughout its circumference, analogous to the hippocampal stratum pyramidale (Brunjes et al., 2005). Thus, it can be assumed that the majority of cells which compose the ‘odour engram’ are excitatory.

- Line 200. The mouse line used does not mark cells activated FOLLOWING 4-OHT injection but all cells that show a CRE Expression at the time of the injections. Taken protein turnover rate into account this can label neurons that were activated many hours before 4-OHT injection. Appropriate controls should be included

Injection with 4-OHT results in Cre-mediated expression within infected cells that were active 1 hour before the injection and up to 4 hours afterwards (Guenthner et al. 2013). To control for unintended labelling, the animals were placed in the behavioural testing room two hours before the time of injection, as described in the methods. Moreover, after testing the animals remained in their homecage for two hours before being transported back to the holding facility. These details are further clarified in the updated manuscript.

2) Line66 “Importantly, control mice similarly trained but tested in a novel context displayed low levels of investigation, indicating context-specificity of the behavioural response”. How can this be? This should be the “baseline” condition and a novel context should induce HIGH levels of investigation...

Behavioural investigation is dependent on the animal experiencing a mismatch between the training context and the lack of an emitted scent from the swab during retrieval. The novel context was not associated with an odour, thus it fails to trigger an odour memory since no mismatch was detected. Under these circumstances, the mouse should show relatively low levels of investigation. Importantly, investigation time measures active sniffing directed towards and within 1 cm of the cotton swab, and does not reflect general exploration, which understandably should be high in novel contexts.

Interestingly, our new hM3D data demonstrated that the artificial activation of tagged-AON neurons does indeed induce high levels of investigation in a novel context. However, it also indicated that the neurons representing the tagged odour memory were not active in control animals.

3) Statistical tests (which one, p values etc.) should be clearly stated in the main text!

The statistical tests employed and their results are presented in the figure captions with their associated data. Including these values in the main text is against Nature Communications formatting.

4) Line 78 “Moreover, c-Fos expression did not differ between groups. This implies that hM4D-mediated inhibition of tagged neurons resulted in the emergence of a parallel network of active neurons which failed to support the odour memory.” This is strange. Why should now a completely different set of cells be active?

We agree with the reviewer that this result is strange and raises an interesting question. It may be explained as a consequence of local circuit interactions, where the loss of function in one cell disinhibits a neighboring cell, ultimately giving rise to a new representation. Furthermore, the emergence of a parallel network of active neurons could support the formation of a new memory, as alluded to by reviewer #3: *“In addition, if the tagged AON neurons indeed represented the conjunctive odor + context experience, does not silencing the tagged AON neurons lead to an entirely new experience when the mice are placed in the otherwise familiar context?”*.

Importantly, this phenomenon is not unique to the AON, but has been observed in other regions of the brain when engrams were artificially silenced (Trouche et al., 2016). Unfortunately, why this occurs and/or whether it occurs naturally is beyond the scope of this investigation.

5) Line 101 “This protocol ensures that internal representations of both chocolate and limonene are formed, but the pattern of activity produced by each odour is tagged separately in designated groups of mice.” Are the amount of labelled cells the same for the two engrams?

Yes, the number of labelled cells is comparable between groups as can be seen in Fig. 3a.

Even if not, additional cells might be tagged that specifically code for the reward and not necessarily for the odor component. Control experiments should be included.

Our manuscript has been updated with a Supplementary Fig. 5 which details the results of our odour memory induction assay in which the mice were trained on chocolate- and limonene-context associations but tagged when subsequently exposed to the training context only. This new control experiment excludes the possibility of neurons specifically coding for the reward from being tagged since the reward stimulus (chocolate) was unavailable at the time of 4-OHT injection.

Our new data demonstrate that when AON neurons were tagged when exposed to chocolate-paired context in the absence of chocolate odour, mice later display robust digging responses upon optogenetic activation. These findings support the presence of specific odour-context representations within the AON.

Other minor points:

- What type of control were used in the study? Reporter only injected animal (line 56) or vehicle-treated animals (line 58)

Multiple controls were used in our study. For example, when examining the efficacy of our tagging method, we compared animals injected with 4-OHT to those injected with the vehicle only (corn oil). In our hM4D and hM3D experiments, we used reporter-only animals to compare the effects of CNO. In our optogenetic experiments, we used different light wavelengths, reporter-only animals, and those tagged with neutral odours as controls.

- The optical stimulation protocol should be better described. E.g. what about the stim strength; was is homogenous between animal? At what stimulation strength was an effect observed. Was the 4 Hz protocol tested with higher stim strength? Was there any correction for the duty cycle etc...

The stimulation protocol has been elaborated on in the updated manuscript.

- “ecphory” should be better explained (Line 142)

This is now explained in the updated manuscript.

- How high was the dropout rate for the 4-OHT injected animals

We did not exclude any animals from our experiments and analyses.

- The injection plane (Bregma +3.0) is close to the OB and other olfactory structures. Retrograde infection and the possibility of antidromic stimulation should be evaluated (e.g. by showing histological sections from the OB, piriform, hippocampus).

While the retrograde infection of AAV is a potential concern, virtually no AAV-mediated expression of the opsin or DREADD genes was observed in the piriform cortex and the hippocampus. Similarly, we did not detect viral expression in the main olfactory bulb, except in its most posterior tip adjacent laterally to the medial AON where we observed only a very sparse labeling. Of note, optic fiber tips implanted over the medial AON were positioned approximately 1 mm lateral to the posterior tip of the OB. Based on our estimation using the light transmission calculator (<https://web.stanford.edu/group/dlab/cgi-bin/graph/chart.php>), the predicted light intensities at the OB and are approximately 0.08 % of original intensity; this is probably an overestimation because light propagates from fiber tip more in the ventral direction than in the lateral direction (see Yizhar et al. 2011 for more detail). We can certainly include the representative images showing the lack of signals (as a supplementary figure) if the reviewers deem it necessary. However, we believe it can simply be stated in the result or method section.

- Line 229: why is there no control for the third experimental group?

Given that limonene is a neutral odour that does not naturally evoke digging, the use of Li-tagged ChETA mice allowed us to evaluate the specificity of the AON engram to the chocolate odour that was experienced at tagging (since it is possible that the stimulation of any subset of neurons within the AON may lead to foraging behaviour). Moreover, the use of Ch-tagged eYFP animals allowed us to control for any influences the chocolate tagging process and subsequent light manipulations may have on inducing digging. Thus, we did not prepare a limonene-tagged eYFP group because we did not deem it to be necessary considering no major confounds exist that are not controlled for by the Li-tagged ChETA and Ch-tagged eYFP groups.

Response to Reviewer #3

In this manuscript, Aqrabawi and Kim address the intriguing question: does the anterior olfactory nucleus (AON) contain odor-specific engrams. Specifically, they address whether the experience of encountering an odor in a specific environment (i.e., conjunctive experience of specific odor + specific context) is represented in AON. They address this by silencing, as well as optogenetically activating, medial portions of AON neurons that are tagged in an activity-dependent (c-Fos-driven) manner, with the aim to test if AON is necessary and sufficient for reactivating context-dependent olfactory experiences. They do so using two behavioral paradigms: (1) the investigative behavior when an odor is removed from a familiar context, and (2) comparison of food-seeking behavior in the presence of chocolate odor in one context vs. in the presence of less appetitive odor in a second context. They use the paradigm (1) for testing the necessity and paradigm (2) for testing the sufficiency.

The question is interesting and an important one. In general, the technique of tagging neurons using the c-Fos pathway seems to work, as many others have shown before for other brain regions. The major concern I have regarding this work is the design and rationales of the behavioral experiments. As explained below, the paradigms used cannot directly distinguish whether the tagged neurons participate in the associative memory (odor + context) or just the olfactory aspects. It is made even more difficult by the fact that rationales for the experimental designs are not clearly explained. As such, in my opinion, with various alternative hypotheses that remain, the data presented currently are, unfortunately, weak and do not necessarily support the claim that AON neurons represent olfactory experiences in specific contexts.

Major concerns:

1) While the authors claim that the tagged neurons represent associative experience of odor + context, it is difficult to distinguish if the neurons represent odor only or odor + context. This is especially the case for the experiment presented in Figure 2. In this dataset, optogenetic activation of tagged neurons leads to food-seeking behavior only when the tagging occurred while mice were exposed to chocolate odor in a specific environment. However, it is also very likely that the chocolate odor on its own would induce this behavior once the conjunction [chocolate odor -> chocolate availability] has been learned. That is, the data could easily be explained as the tagged AON neurons representing the olfactory stimuli, which could then drive food-seeking behavior downstream. A more convincing experiment would be to tag AON neurons when only the context “A” is presented. The supplementary figure 2 shows that mere exposure to the context “A” leads to reactivation of the memory and thus food-seeking behavior. If tagging AON neurons in the absence of odor this way leads to food-seeking behavior, it would dissociate the two alternative hypotheses (AON representing odor only vs. odor + context) and be a convincing result indeed.

We thank the reviewer for their critical insight. The updated manuscript has been significantly improved with the addition of our new data presented as Supplementary Fig. 5 which details the results of the experiment suggested. Mice underwent similar odour-context association training to those described in Fig. 2, yet AON neurons were tagged during the presentation of context “A” (or “B”) only. Under these

conditions, mice tagged in the chocolate-associated context only still exhibited robust digging in response to optogenetic activation of the tagged neurons. This finding supports the idea that the conjunctive odour-context experience is represented in the AON, and that context alone can cue reinstatement of the odour memory.

2) As for the first behavioral paradigm presented in Fig 1, this seems to be assessing novelty detection. The rationale for using this paradigm, in particular, an explanation on what is driving the mice to investigate the cotton swab, is not given clearly. Is it the absence of an odor when it is expected, or is it that the swab on its own smells different, and as a result regarded as a novel object?

The reviewer has rightfully pointed out the need for further clarification on the use of the paradigm adopted for Fig 1. We have now expanded our rationale and explanations in the updated manuscript.

3) In addition, if the tagged AON neurons indeed represented the conjunctive odor + context experience, does not silencing the tagged AON neurons lead to an entirely new experience when the mice are placed in the otherwise familiar context?

Indeed, this may be the reason why we see the emergence of a parallel network of active neurons in the AON upon silencing the tagged population (Fig. 1f). However, whether this reflects the formation of an entirely new memory when mice are placed in the familiar context is difficult to determine.

4) The manuscript presents only the time spent investigating the cotton swabs. As a result, it is difficult to understand what is going on. A crucial control that is missing here, in order to show an importance of context in the reactivated AON neurons, is an assessment of the overlap in cFos-expressing population and the mCherry-expressing population when mice are exposed to context “B” AND the familiar odor. This needs to be contrasted to the proportion of overlap when the “full” stimulus is given (context “A” + familiar odor). These data would give an estimate on the extent to which re-activation depends on odor only vs. odor + context.

We attempted to demonstrate the importance of context for reactivating AON neurons in our hM4D experiments. We have shown that exposure to the training context “A” only was sufficient to increase the degree to which the original tagged population was reactivated, compared to animals exposed to a novel context (Fig. 1f). Thus, specific contextual information certainly contributes to the faithful reinstatement of patterns of activity represented in the AON. However, the reviewer raised an interesting and important concern. The contextual information may be used to reactivate neurons reflective of the odour only and may not itself be embedded as part of the pattern of activity.

Thus, we conducted an additional control, depicted as Supplementary Fig. 2, to examine the dependence of AON patterns of activity on context. We compared the degree of reactivation in mice exposed to the context “A” + familiar odour with those exposed to a novel context B + familiar odour. While levels of c-Fos+ cells were comparable between groups, animals exposed to a novel context B + familiar odour displayed significantly lower rates of reactivation of the original tagged population, despite being presented with the same odour. This assessment clearly demonstrates a contribution of context in forming the framework of activity within the AON.

5) The authors used two different behaviors to test the necessity and sufficiency – that is, the necessity for one behavior, and sufficiency for a different behavior. The first behavior assesses novelty detection, and the second, food-seeking behaviors. It is unclear why the authors did this.

Our updated manuscript now includes the results of a gain-of-function experiment using the paradigm described in Fig. 1. This additional data improves the symmetry in our approach since the same behaviour was examined under both inhibition and activation of tagged AON neurons. While the familiarity-based memory task is widely used as an effective test of olfactory memory, it is intrinsically limited because it does not directly measure memory contents. To gain more direct behavioural evidence that specifically addresses the content of AON engrams, we designed a protocol that exploits the animal's goal-directed, foraging behaviour. We have made our choice of behavioural paradigms clear in the updated manuscript.

6) Regarding the contexts: are the contextual cues represented in the AON visual, or olfactory? In order to distinguish the two, more clean way to change the visual context would be to create different contexts “virtually” – i.e., using the same cage and contents, but by projecting different images. If this is too difficult, the authors should at least detail how the cages had been cleaned and describe in more detail what parameters should differ between the two contexts.

We thank the reviewer for pointing out the need for further clarification on our contextual configurations. Although we have briefly outlined our procedure for creating individual contexts within the method sections corresponding to the behavioural tests, we have added an additional note which reiterates our general approach.

We have used two methods for creating the different contexts in our paradigms. In each case, transparent cages of the same size were used. One method involved pasting visually patterned wallpaper on the exterior walls. Wallpapers which depict different designs represented different ‘visual’ contexts. Another method was to use transparent cages with different colourful objects (plastic toys, coffee cups, etc) placed around the outside of the cage. Our real-time place preference test had the additional factor of metal fretwork flooring that differed between the two contexts.

Other minor comments:

1) Choi et al in 2011 (DOI 10.1016/j.cell.2011.07.041) previously showed that activity-dependent tagging of piriform neurons can drive opposing behaviors depending on the experience. Given that AON and the piriform cortex have dense interconnections, what distinct roles do authors think that the two regions have? In addition, the authors need to cite the above paper.

The updated manuscript includes an expanded discussion which cites Choi et al (2011) and describes the potential contributions of the piriform cortex in mediating odour memory expression.

2) Regarding the data presented in Fig. 1f, the authors state that the data come from “a subset of mice” (line 75). Can the authors please elaborate?

Our hM4D experiments involved the use of 10 mice/group. However, in analyzing the correlation between the reactivation rate and investigation time we have used the data from 6 mice/group that were randomly chosen for histological purposes.

3) In general, please describe the experiments more clearly and with more detail. For example, in Fig 1, when the mice are exposed to context B, was the familiar odor present? Were control mice given CNO (line 66)?

The manuscript has been updated to make these details clear.

4) Line 59: Please explain the enrichment.

We have now elaborated on the enrichment in the updated manuscript.

5) Line 62: Please explain how much later (e.g. hours or days).

This detail has now been added to the updated manuscript.

6) Line 66: Can it be that the memory is made inaccessible, instead of erased?

We have replaced the word 'erased' with 'silenced' to more accurately describe our interpretation.

7) Line 105: Please elaborate on the stimulation parameters, in particular stimulation duration.

The stimulation duration and pulse width for the optogenetic activation is now included in the methods section.

Reviewers' Comments:

Reviewer #1:

Remarks to the Author:

The authors have addressed several of the points raised by the reviewers and the manuscript has been strengthened. However, the issue of what exactly is contributed by AON as revealed by the cFos+ ensemble experiments (odor versus odor+context) remains somewhat unclear. An unambiguous clarification is essential for this study to provide a firm advance over previous ones.

I would recommend that the authors provide at least two of the additional sets of data required upon the first reviewing process:

- 1) the positive controls in pyriform cortex (3rd point, Rev2) should hopefully provide evidence as to how cFos+ cells in AON contribute to behaviour in a way distinct from primarily reflecting odor identity
- 2) tagging at the beginning of the task (1 day of training), followed by early (as opposed to late) interference experiments (second point, Rev1) in order to determine whether the AON ensembles might be specifically important for odor-context retrieval, as opposed to mismatch from robustly trained odor-context expectations.

Reviewer #2:

Remarks to the Author:

The manuscript generally improved but major points concerning the interpretation of the activity dependent labelling remain unanswered:

- 1) The way the data are currently presented convey the impression that odour only (or context only) presentation are not sufficient to increase AON activity beyond baseline (that is activity measured in terms of c-Fos level expression in the AON). However, what does this really mean? As the author's state: "our c-Fos counting data likely underestimates the number of neurons that are activated by different behavioural conditions" it might be that they are simply detecting tip-of-the-iceberg activity. Like suggested already, showing c-Fos labelled cells in the pyriform cortex after odor exposure would be important in this respect. If there is no, or only weak labelling in PIR after odor exposure under their conditions (homeage, odor exposure) this would show that this is NOT a special AON feature but rather a limitation of their detection method. In this case it should not be stated in absolute terms that odor and context, only when concurrently presented increase AON activity since this conveys the (very likely wrong) impression that there are no odor-only AON responsive cells.
- 2) While this reviewer appreciates the additionally preformed experiments the interpretation of the data is more than challenging. In Supp Fig 5 the authors tag neurons when exposed to the training context only, however FOLLOWING training. The only explanation for these positive outcome (mice displaying robust digging responses upon stimulation) is that when this context was presented in isolation, this lead to a reactivation of odor+context cells because the association has been trained for days. This is also what the authors report: "We have shown that exposure to the training context "A" only was sufficient to increase the degree to which the original tagged population was reactivated, compared to animals exposed to a novel context (Fig. 1f)." So, in contrast to what the authors claim, these data do not allow to make assumptions if AON activity encodes for e.g. reward or like questioned by Reviewer 3 to differentiate between odor or context. The authors should have labeled context only AON cells before training. Then, it would have been informative to show that reactivation of only the context cells after learning is sufficient to elicit digging behavior.

3) Novelty vs. salient stimulus vs. mismatch. The authors claim that novelty is not required for AON activity and that basically everything outside the homecage can serve as a "salient" stimulus that when paired with an odorant will lead to AON activation. This logic is very hard to follow. E.g. it is hard to understand why at the fifth exposure to a spatial context + odor this context still can serve a "salient" stimulus to label AON cells. Also the mismatch interpretation that the authors use is tricky since clearly no mismatch should be detected by the animal once it learns the odor context association. Moreover this effect should wear of over time.

4) Another previous question was not fully answered: what about the stim strength; was is homogenous between animal? At what stimulation strength was an effect observed. Was the 4 Hz protocol tested with higher stim strength? Was there any correction for the duty cycle

Reviewer #3:

Remarks to the Author:

The revision has improved the manuscript.

In the future, please detail locations (pages + lines) of edited texts when these form your responses to reviewer's comments. Present lack of this information has made the evaluation very difficult and can appear, even if unintended, to be evasive.

Reviewer #1 (Remarks to the Author):

The authors have addressed several of the points raised by the reviewers and the manuscript has been strengthened. However, the issue of what exactly is contributed by AON as revealed by the cFos+ ensemble experiments (odor versus odor+context) remains somewhat unclear. An unambiguous clarification is essential for this study to provide a firm advance over previous ones.

I would recommend that the authors provide at least two of the additional sets of data required upon the first reviewing process:

1) the positive controls in piriform cortex (3rd point, Rev2) should hopefully provide evidence as to how cFos+ cells in AON contribute to behaviour in a way distinct from primarily reflecting odor identity

The manuscript now features an updated Supplementary Fig. 1 which compares c-Fos expression in the homecage to the density of c-Fos⁺ cells in the AON and anterior piriform cortex under ‘odour-only’ and ‘odour-context’ conditions. The results obtained for c-Fos expression in the AON is consistent with earlier work (Aqrabawi & Kim, 2018; Kay et al., 2011) and the density measures obtained in our tagging experiments. The AON exhibited a widespread increase in c-Fos⁺ cells/mm², relative to homecage levels, when the animal is exposed to coincident olfactory and contextual information, but not when salient contextual input is absent, as in the odour-only condition. Distinctively, the piriform cortex shows an increase in c-Fos expression when the odour is presented, regardless of whether this occurred in the homecage or in a salient context. Thus, it appears bottom-up sensory input alone is sufficient to drive piriform activity, but this is not the case for the AON. The necessity of contextually-relevant stimuli for AON activation is evidence of a unique role in contextual processing, suggesting a function beyond simply coding/relaying odour identity. Indeed, the distinct contribution of context to AON activity is also clearly highlighted in our Supplementary Fig. 2 where the same olfactory stimulus within different contexts was shown to result in significantly disparate representations within the AON.

2) tagging at the beginning of the task (1 day of training), followed by early (as opposed to late) interference experiments (second point, Rev1) in order to determine whether the AON ensembles might be specifically important for odor-context retrieval, as opposed to mismatch from robustly trained odor-context expectations.

We understand the reviewer’s concern and agree with the need to disambiguate between the two functions. To examine whether AON neurons encode single-trial, odour-context learning events in support of odour memory, we tagged AON neurons with ChETA-eYFP during the initial exposure to chocolate within a salient context and subsequently stimulated the representation after 7 days with no further interim training. Despite the fact that the mice were not repeatedly exposed to the chocolate/context pairing, ChETA-eYFP-tagged animals demonstrated digging behaviour under blue light illumination compared to reporter controls. Yet, in this test the extent of their digging is notably lower than ChETA-eYFP-expressing animals who have received extensive training before tagging the chocolate odour. Nonetheless, this data provides evidence that the AON is capable of forming odour-context representations following only one exposure which can later be retrieved to drive relevant behaviour.

Reviewer #2 (Remarks to the Author):

The manuscript generally improved but major points concerning the interpretation of the activity dependent labelling remain unanswered:

1) The way the data are currently presented convey the impression that odour only (or context only) presentation are not sufficient to increase AON activity beyond baseline (that is activity measured in terms of c-Fos level expression in the AON). However, what does this really mean? As the author's state: "our c-Fos counting data likely underestimates the number of neurons that are activated by different behavioural conditions" it might be that they are simply detecting tip-of-the-iceberg activity. Like suggested already, showing c-Fos labelled cells in the piriform cortex after odor exposure would be important in this respect. If there is no, or only weak labelling in PIR after odor exposure under their conditions (homecage, odor exposure) this would show that this is NOT a special AON feature but rather a limitation of their detection method. In this case it should not be stated in absolute terms that odor and context, only when concurrently presented increase AON activity since this conveys the (very likely wrong) impression that there are no odor-only AON responsive cells.

To demonstrate that the coincidence detection observed in the AON is not a limitation of our detection method, but a distinct property of the AON, we followed the reviewer's suggestion, examining piriform cortical c-Fos expression in response to odour-only and odour-context conditions. As expected, we found that the piriform cortex exhibits an increase in c-Fos expression in both conditions, regardless of the contextual relevance of the odour. In contrast, the AON only displayed a greater c-Fos density under the conjunctive exposure to both olfactory and contextual stimuli. We assert again, this does not imply that the AON does not contain odour-only AON responsive cells, or even that it is void of c-Fos in the odour-only condition; we are simply demonstrating the existence of a substantial number of neurons within the structure that do require coincident odour-context input to express c-Fos.

2) While this reviewer appreciates the additionally preformed experiments the interpretation of the data is more than challenging. In Supp Fig 5 the authors tag neurons when exposed to the training context only, however FOLLOWING training. The only explanation for these positive outcome (mice displaying robust digging responses upon stimulation) is that when this context was presented in isolation, this lead to a reactivation of odor+context cells because the association has been trained for days. This is also what the authors report: "We have shown that exposure to the training context "A" only was sufficient to increase the degree to which the original tagged population was reactivated, compared to animals exposed to a novel context (Fig. 1f)." So, in contrast to what the authors claim, these data do not allow to make assumptions if AON activity encodes for e.g. reward or like questioned by Reviewer 3 to differentiate between odor or context. The authors should have labeled context only AON cells before training. Then, it would have been informative to show that reactivation of only the context cells after learning is sufficient to elicit digging behavior.

Conducting this experiment would unlikely yield any meaningful outcomes. Please note that the context alone is sufficient to increase AON activity only after (not before) the context has been associated with an odour. Thus, before training, there are hardly any 'context only' AON cells. Assuming that we are capable of tagging a very few neurons following exposure to the 'context-only' condition, it would be unreasonable to believe that the animals would forage upon stimulating such a small population of cells.

3) Novelty vs. salient stimulus vs. mismatch. The authors claim that novelty is not required for AON activity and that basically everything outside the homecage can serve as a “salient” stimulus that when paired with an odorant will lead to AON activation. This logic is very hard to follow. E.g. it is hard to understand why at the fifth exposure to a spatial context + odor this context still can serve a “salient” stimulus to label AON cells.

Context is widely understood in the memory literature to be a powerful retrieval cue. Contextual input alone is sufficient to retrieve sensory-rich memories well beyond initial encoding (Nyberg et al., 2000; Gottfried et al., 2004; Mandairon et al., 2014; Tanaka et al., 2014; Aqrabawi and Kim, 2018). Thus, the role of context is not limited to the initial formation of episodic memories, but it continuously plays a critical function in retrieval as well. Thus, even at the fifth exposure to an odour-context pairing, both simultaneously presented cues remain salient for driving AON activity, for encoding a representation specific to that encounter and for retrieving information related to previous encounters. Nonetheless, our data demonstrating significantly elevated c-Fos expression in the AON at the fifth, and at even later exposures, provides strong evidence that contextual salience persists.

Also the mismatch interpretation that the authors use is tricky since clearly no mismatch should be detected by the animal once it learns the odor context association. Moreover this effect should wear off over time.

It is difficult to understand the reviewer’s logic. Once the animal learns an association, any relevant input that differs from this established association is considered a mismatch. For example, if the animal learns that *odour 1=context A*, all inputs containing the element ‘odour 1’ or ‘context A’ that is not faithful to the original representation is considered a mismatch (e.g. *odour 1=context B*; *context A=odour 2*, or as we have done in our memory test, *context A=no odour*). In fact, no mismatch should be detected if the animal has not learned an association, since there is no representation to match with.

4) Another previous question was not fully answered: what about the stim strength; was it homogenous between animal? At what stimulation strength was an effect observed. Was the 4 Hz protocol tested with higher stim strength? Was there any correction for the duty cycle?

We have indicated that a 2 mW laser power was used for stimulation. This value was measured using a digital power meter (Thorlabs) from light emitted through each optic fiber before implantation. The actual light power was adjusted for the animals so that each mouse effectively received a 2 mW stimulation strength. This power was maintained for all wavelengths and frequencies used in our study. We did not manually make any correction for the duty cycle, our waveform generator (Agilent, 33220A) was used to maintain a pulse width of 10 ms.

Reviewer #3 (Remarks to the Author):

The revision has improved the manuscript.

In the future, please detail locations (pages + lines) of edited texts when these form your responses to reviewer's comments. Present lack of this information has made the evaluation very difficult and can appear, even if unintended, to be evasive.

We thank the reviewer for their service. In particular, the insightful suggestions which have undoubtedly strengthened our work.

Reviewers' Comments:

Reviewer #1:

Remarks to the Author:

The authors have carefully addressed my remaining points, and the additional data are very informative. I think that in its present form the manuscript provides an important advance that will be of substantial interest to neuroscientists.

Reviewer #2:

Remarks to the Author:

Thought the added experiment somewhat improved the MS I still have several concerns:

- 1) Like mentioned in the rebuttal it should be also clearly stated in the main MS that the findings do not imply that the AON does not contain odor-only AON responsive cells.
- 2) Effectivity of activity dependent labelling approaches vary between brain areas and cell types; also that should be mentioned / discussed in the MS.
- 3) Concerning data interpretation:

"Conducting this experiment would unlikely yield any meaningful outcomes. Please note that the context alone is sufficient to increase AON activity only after (not before) the context has been associated with an odour. Thus, before training, there are hardly any 'context only' AON cells. Assuming that we are capable of tagging a very few neurons following exposure to the 'context-only' condition, it would be unreasonable to believe that the animals would forage upon stimulating such a small population of cells."

The reviewer acknowledges that the proposed experiment might be technically challenging. However, without these or additional experiments, in the current form of the MS, several conclusion are invalid:

"The lack of an explicit response in limonene-tagged mice further supports the existence of odour specific engrams stored within the AON. Though, the possibility remains that additional cells specifically coding for the rewarding aspects of chocolate were involved in the tagged population. As a control, we repeated our odour memory induction assay, yet AON neurons were tagged during presentation of the odour-associated contexts only"

- The experiments do not show that the tagged cells represent an odor specific engram. Instead odor-context or odor-reward engrams or combinations of them cannot be excluded because performing tagging after training is a *Contradictio in adiecto*; the results can't be conclusive. If the authors are not able to design an experiments underlying their hypothesis, e.g. applying only "chocolate-smell" to avoid physical reward associations in the same context and tagging these neurons, this part should be toned down or removed completely.

"It is difficult to understand the reviewer's logic. Once the animal learns an association, any relevant input that differs from this established association is considered a mismatch. For example, if the animal learns that odour 1=context A, all inputs containing the element 'odour 1' or 'context A' that is not faithful to the original representation is considered a mismatch (e.g. odour 1=context B; context A=odour 2, or as we have done in our memory test, context A=no odour). In fact, no mismatch should be detected if the animal has not learned an association, since there is no representation to match with."

- What is your definition of mismatch? This reviewer believes that the authors are using mismatch detection to explain the activity dependent labelling observed in the odor-context association. However, no mismatch should be detected by the animal once it learns the odor-context association, which should translate to no c-fos expression in later exposures. Since this is not the case in the reported experiments, the mismatch detection theory is very likely the wrong interpretation. If this is not what the authors wanted to say, they should be more explicit about their idea of AON mismatch

detection.

"We have indicated that a 2 mW laser power was used for stimulation. This value was measured using a digital power meter (Thorlabs) from light emitted through each optic fiber before implantation. The actual light power was adjusted for the animals so that each mouse effectively received a 2 mW stimulation strength. This power was maintained for all wavelengths and frequencies used in our study. We did not manually make any correction for the duty cycle, our waveform generator (Agilent, 33220A) was used to maintain a pulse width of 10 ms."

It is difficult to understand the logic: how can a 2mW stimulation strength be maintained for all frequencies if no correction for the duty cycle have been made?

Response to Reviewers' Comments

Reviewer #1 (Remarks to the Author):

The authors have carefully addressed my remaining points, and the additional data are very informative. I think that in its present form the manuscript provides an important advance that will be of substantial interest to neuroscientists.

We thank the Reviewer for the service, in particular the insightful suggestions which have undoubtedly strengthened our work.

Reviewer #2 (Remarks to the Author):

Thought the added experiment somewhat improved the MS I still have several concerns:

- 1) Like mentioned in the rebuttal it should be also clearly stated in the main MS that the findings do not imply that the AON does not contain odor-only AON responsive cells.**
- 2) Effectivity of activity dependent labelling approaches vary between brain areas and cell types; also that should be mentioned / discussed in the MS.**

We have now updated the manuscript to reflect the Reviewer's point (lines 53-57).

3) Concerning data interpretation:

“Conducting this experiment would unlikely yield any meaningful outcomes. Please note that the context alone is sufficient to increase AON activity only after (not before) the context has been associated with an odour. Thus, before training, there are hardly any ‘context only’ AON cells. Assuming that we are capable of tagging a very few neurons following exposure to the ‘context-only’ condition, it would be unreasonable to believe that the animals would forage upon stimulating such a small population of cells.”

The reviewer acknowledges that the proposed experiment might be technically challenging. However, without these or additional experiments, in the current form of the MS, several conclusions are invalid:

“The lack of an explicit response in limonene-tagged mice further supports the existence of odour specific engrams stored within the AON. Though, the possibility remains that additional cells specifically coding for the rewarding aspects of chocolate were involved in the tagged population. As a control, we repeated our odour memory induction assay, yet AON neurons were tagged during presentation of the odour-associated contexts only”

- The experiments do not show that the tagged cells represent an odor specific engram. Instead odor-context or odor-reward engrams or combinations of them cannot be excluded because performing tagging after training is a Contradictio in adiecto; the results can't be conclusive. If the authors are not able to design experiments underlying their hypothesis, e.g. applying only “chocolate-smell” to avoid physical reward associations in the same context and tagging these neurons, this part should be toned down or removed completely.

Our study does not claim that the tagged cells exclusively represent the olfactory stimulus. In fact, we have explicitly demonstrated the influence of contextual information in forming the representation within the AON. Supplementary Fig. 2 examined the dependence of AON activity patterns on context. All animals were tagged during an initial exposure to an odour + context “A”. We compared the degree of reactivation in mice re-exposed to the context A + odour to those that are instead exposed to

a novel context B + the same odour. While levels of c-Fos+ cells were comparable between groups, animals exposed to a novel context B + familiar odour displayed significantly lower rates of reactivation of the original tagged population, despite being presented with the same odour. This assessment clearly demonstrates a contribution of context in forming the framework of activity within the AON. Thus, in the Reviewer's words "the experiments do not show that the tagged cells represent an odor-specific engram", but an episodic memory trace of the odor and its associated context.

As an alternative to the reviewer #1's suggestion and our new data (Figures 5 and 6), the reviewer suggested the following experiment to avoid the above-mentioned issue of physical reward associations:

'e.g., applying only "chocolate-smell" to avoid physical reward associations in the same context and tagging these neurons'.

Please note that even the application of only "chocolate-smell" cannot completely exclude the physical reward association because the chocolate-smell itself may carry an innate rewarding component. How can one be completely sure that an olfactory experience is purely olfactory, not accompanied by any emotional value? We clearly acknowledged this in our manuscript by stating that "the possibility remains that additional cells specifically coding for the rewarding aspects of chocolate were involved in the tagged population".

"It is difficult to understand the reviewer's logic. Once the animal learns an association, any relevant input that differs from this established association is considered a mismatch. For example, if the animal learns that odour 1=context A, all inputs containing the element 'odour 1' or 'context A' that is not faithful to the original representation is considered a mismatch (e.g. odour 1=context B; context A=odour 2, or as we have done in our memory test, context A=no odour). In fact, no mismatch should be detected if the animal has not learned an association, since there is no representation to match with."

- What is your definition of mismatch? This reviewer believes that the authors are using mismatch detection to explain the activity dependent labelling observed in the odor-context association. However, no mismatch should be detected by the animal once it learns the odor-context association, which should translate to no c-fos expression in later exposures. Since this is not the case in the reported experiments, the mismatch detection theory is very likely the wrong interpretation. If this is not what the authors wanted to say, they should be more explicit about their idea of AON mismatch detection.

We have defined 'mismatch' in the text quoted by the Reviewer. Furthermore, we are not using mismatch to explain the activity-dependent labelling observed in the odour-context associations. Instead, we postulate 1) that an increase in AON activity is the consequence of coincidence detection of salient olfactory and contextual input, and 2) that the AON activity likely contributes to mismatch detection between experience and expectation. If the Reviewer had proposed any reasonable alternatives to these as explanation, we would have been delighted to consider and discuss such possibilities in our manuscript.

"We have indicated that a 2 mW laser power was used for stimulation. This value was measured using a digital power meter (Thorlabs) from light emitted through each optic fiber before implantation. The actual light power was adjusted for the animals so that each mouse effectively received a 2 mW stimulation strength. This power was maintained for all wavelengths and

frequencies used in our study. We did not manually make any correction for the duty cycle, our waveform generator (Agilent, 33220A) was used to maintain a pulse width of 10 ms." It is difficult to understand the logic: how can a 2mW stimulation strength be maintained for all frequencies if no correction for the duty cycle have been made?

We did not manually make any correction for the duty cycle, our waveform generator (Agilent, 33220A) automatically adjusts the duty cycle when the output frequency is changed. The light power was measured using a digital power meter for each frequency to confirm a constant 2 mW output. For further information, please refer to page 57 of the user manual (download link: <https://web.sonoma.edu/ese/manuals/33220-90002.pdf>).